

# Enhancing the accessibility of unified modeling systems: GFDL SHiELD v2021b in a container

Kai-Yuan Cheng[1,2], Lucas M. Harris[2], Yong Qiang Sun[1,2,3]

[1]Cooperative Institute for Modeling the Earth System, Program in Oceanic and Atmospheric Sciences, Princeton University, Princeton, NJ, USA
[2]NOAA Geophysical Fluid Dynamics Laboratory, Princeton, NJ, USA
[3]Now at Department of Earth, Environmental and Planetary Sciences, Rice University, Houston, TX, USA

*Correspondence to*: Kai-Yuan Cheng (kai-yaun.cheng@noaa.gov)

**Abstract.** Container technology provides a pathway to facilitate easy access to unified modeling systems and opens opportunities for collaborative model development and interactive learning. In this paper, we present the implementation of software containers for the System for High-resolution prediction on Earth-to-Local Domains (SHiELD), a unified atmospheric model for weather-to-seasonal prediction. The containerized SHiELD is cross-platform and easy to install. Flexibility of the containerized SHiELD is demonstrated as it can be configured as a global, a global-nest, and a regional model. Bitwise reproducibility is achieved on various x86 systems tested in this study. Performance and scalability of the containerized SHiELD are evaluated and discussed.

## 1 Introduction

Unified modeling systems have shown advantages as a single framework supporting versatile applications across a wide range of spatial and temporal scales. Such a system helps accelerate model development as it provides an ideal platform for modelers/scientists to gather together and combine efforts to improve the model. The Unified Model of the U.K. Met Office, the most notable unified modeling system, adopted the unified modeling approach and demonstrated its strength in terms of integration and collaboration (Walters et al., 2011). In addition to facilitating collaboration, the system opens the possibility of developing one model with multiple uses. Previous studies (e.g., Brown et al., 2012 and Harris et al., 2020) have shown that it is possible to develop multiple applications at the same time for multiple purposes, such as conducting severe weather forecast and climate prediction simultaneously. Finally, the unified modeling system allows users to apply lessons learned from one application to another application. For example, Brown et al. (2012) showed that the error growth calculated based on short-range predictions can be used to evaluate the performance of climate predictions. Likewise, climatological signals deduced from the model provide valuable information for the development of physics parameterizations that can be used in short-range weather forecasts.

Lowering the barriers of entry into unified modeling systems will open many opportunities for the earth science community. To that end, the Unified Forecast System has been making consistent steps to make the system accessible and encourage discussion and collaborative research, one example being the Graduate Student Test



(https://ufscommunity.org/science/gst/, last access: 13 July 2021). As discussed, a unified infrastructure provides a bedrock for cooperation. Assuring easy access to the infrastructure will help achieve cooperation. In an educational setting, Hacker et al. (2017) has shown that better access to a mesoscale model benefits classroom learning. Better access to unified modeling
systems can greatly improve learning experience. For example, a highly-configurable unified model can be used as a global model to demonstrate the concept of climate sensitivity, as a regional model to learn the impact of topography on the track of a hurricane, or, as a tool to study multiscale interactions.

A software container provides a pathway to enhance the accessibility of geoscientific models. A software container, or simply container, is a stand-alone, executable software that is designed to deploy applications with portability and
performance. There are a few geoscientific models being implemented in software containers. Hacker et al. (2017) containerized the Weather Research and Forecasting model (WRF), a regional atmospheric model. Melton et al. (2020) containerized the Canadian Land Surface Scheme including Biogeochemical Cycles (CLASSIC). Jung et al. (2021) containerized the regional ocean-modeling system (ROMS). However, due to the fact that unified modeling systems are complicated and traditionally developed on specialized machines (e.g., high-performance computers), there appears to be no
unified modeling system being containerized yet. This paper attempts to take advantage of container technology and make unified modeling systems approachable.

The purpose of this paper is to describe the implementation of software containers for the System for High-resolution prediction on Earth-to-Local Domains (SHiELD), a compact unified atmospheric model developed at the Geophysical Fluid Dynamics Laboratory (GFDL). Simulations of Hurricane Laura, with a regional and a global-nest configuration, are
conducted on different computer systems to demonstrate the flexibility, portability, and easy use of the containerized SHiELD. Performance and scalability of the containerized SHiELD are examined and discussed. Future work and potential applications are discussed.

## 2 SHiELD in a container

### 2.1 SHiELD

The System for High-resolution prediction on Earth-to-Local Domains (SHiELD), a unified atmospheric model developed at the GFDL, has demonstrated its capability for versatile applications on a wide range of temporal and spatial scales, including severe weather nowcasting, hurricane forecasting, and subseasonal-to-seasonal prediction (Harris et al., 2020). SHiELD is powered by the finite-volume cubed-sphere dynamical core (FV3; Putman and Lin, 2007; Harris and Lin, 2013) and is equipped with a modified version of the Global Forecast System (GFS) physics suite developed by the Environmental
Modeling Center of the National Centers for Environmental Prediction. As a unified modeling system, SHiELD has been used for forecast, research, and model development: all in a single framework (Harris et al., 2020). For example, SHiELD, featuring variable-resolution, has demonstrated excellence in tropical cyclone forecasting (Hazelton et al., 2018). Another example is that SHiELD was used to develop a time-split microphysics parameterization for multiple applications, such as





convective scale weather prediction and global scale cloud-radiative forcing research (Harris et al., 2020). Version 2021b of
the SHiELD is used in study and the model source code can be found in Cheng et al. (2021).

## 2.2 Containerization

Containerization refers to packaging one or more applications (such as a unified atmospheric model) into a container in a
portable manner. A container packages not only applications but all their dependencies, such as runtime environment and
libraries, so the applications can run directly from one computing system to another. Unlike a virtual machine that emulates a
whole computing system for use at the hardware layer, a container uses the kernel of the host machine and packages only the
necessary components required to run applications. As a result, a container is lightweight and fast.

Advantages of the containerization of geoscience models are discussed in many papers (e.g., Hacker et al., 2017 and
Melton et al., 2020) and will be shown throughout this paper, including easy installation, high portability, and perfect
reproducibility. We want to add one additional advantage that motivates innovation and spurs model development. It is not
uncommon for developers to come up with innovative schemes that are not ready for publication but ready for technology
transfer and/or public use. In this situation, developers may want to take advantage of container technology to package their
innovations as a black box that users can use but cannot see through. Unlike source code sharing which reveals everything,
software containers may be used to protect developers' intellectual properties without revealing their full content while being
used to share with others.

Technically speaking, a container does not directly package applications and dependencies. It is the container image, or
simply image, that does the packaging. An image is an immutable file that contains prebuilt applications and their
dependencies needed to run the applications. An image is used by a container to construct a runtime environment and then
run applications.

For the containerization of SHiELD, we use Docker (https://www.docker.com/, last access: 13 July 2021) as a primary
tool and Singularity (https://sylabs.io/, last access: 13 July 2021) as a secondary. Docker is a leading containerization
platform that sets the industry standard for containers. However, Docker containerization requires superuser access, which is
a concern for multi-user systems like supercomputers. As a result, most supercomputers do not allow Docker to be installed.
Singularity, on the other hand, is designed to address the security concern. Singularity is originally designed for
supercomputers (Kurtzer et al., 2017) and architected specifically for large-scale and performance-oriented applications.
Both Docker and Singularity are available for free.

The procedure of containerizing SHiELD is described as follows. First, we create a Docker image with SHiELD and its
dependencies. In the spirit of open collaboration, the SHiELD image does not contain any proprietary software. SHiELD and
all its dependencies are open-source. SHiELD is built using open-source compilers: GNU Compiler Collection (GCC) and
GFortran. Second, we create a SHiELD container by containerizing the SHiELD image on any supported system. As of July
2021, Docker supports three major operating systems (OS): Windows, macOS, and Linux. The SHiELD image can also be
used in cloud computing. Major cloud computing platforms (e.g., Amazon Web Services and Microsoft Azure) can directly





deploy Docker containers. On supercomputers where Docker is generally not available, the SHiELD image can be easily converted to a Singularity container image and seamlessly executed. One thing to note is that the container made by either Singularity or Docker gives identical results, which will be discussed in the next section.

## 100    3 Running containerized SHiELD as a regional and a global-nest model

For the purpose of demonstration and experimentation, we conducted 24-hour simulations of Hurricane Laura initialized from 2021082612 UTC. Two different domain configurations, as illustrated in Fig. 1, are used to demonstrate the flexibility of SHiELD, showing that it can be used for a variety of applications at different spatial and temporal scales. The first configuration is a regional domain centered over the Gulf Coast of the United States. The domain size is $108 \times 108$ grid cells

with a grid-cell width of approximately 35.5 km. Initial conditions and a time-series of boundary conditions for the regional configuration are generated from the U.S. operational Global Forecast System (GFS) analysis.

     The second configuration is a global domain embedding a locally-refined nest domain centered over the Gulf Coast of the United States. The global domain is a cubed-sphere with $96 \times 96$ grid cells on each of the six tiles, which yields an average grid-cell width of approximately 100 km. The nest domain shares identical size and position with the regional domain. This

global-nest configuration requires only initial conditions from the GFS analysis and enables two-way interaction between the global and the nest domains. The details of the two-way nesting method can be found in Harris and Lin (2013).

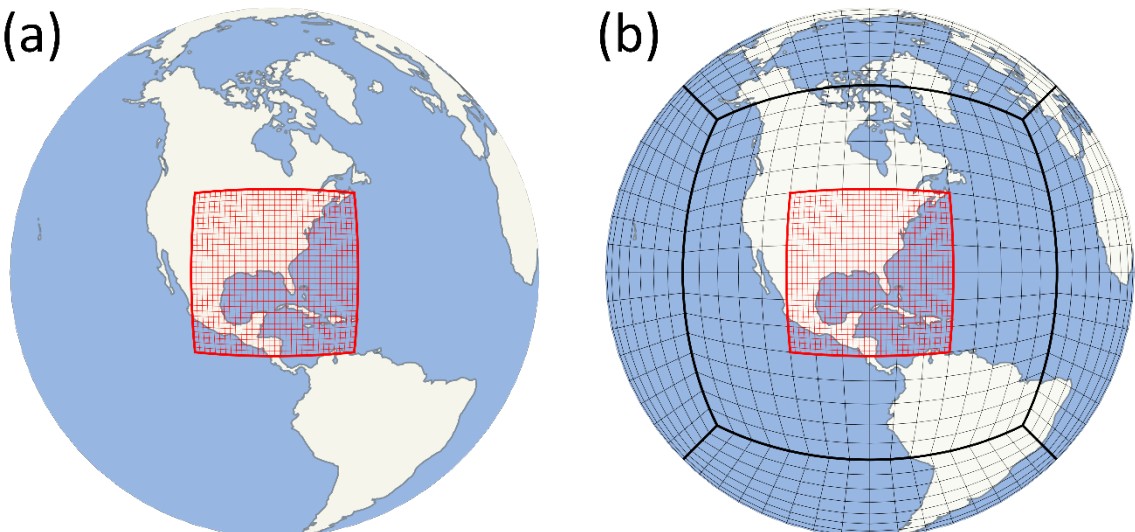

**Figure 1: Two grid configurations. (a) A 37.5 km regional grid (red mesh). (b) A 100 km global grid (black mesh) with a nest grid (red mesh) that is refined by a factor of 3; the position and the grid size of the nest grid are the same as those of the regional grid.**
**Each grid box represents 12×12 actual grid boxes. Black heavy lines are the cubed-sphere edges and red heavy lines are the boundaries of the regional/nest domain.**



Both configurations use identical 63 Lagrangian vertical levels, the same as those used in GFS version 15. The vertical resolution is finest at the bottom level and gradually expands with height, as shown in Fig. 2. As for timestep, both regional and global-nest runs use the same time step (450 s) for physical parameterization. The regional run uses an acoustic timestep
of 75 s. For the global-nest run, the acoustic time step is 150 s and 75 s in the global and the nest grid, respectively. The resolution, timesteps, and physical parameterizations of the regional and the global-nest run are summarized in Table 1.

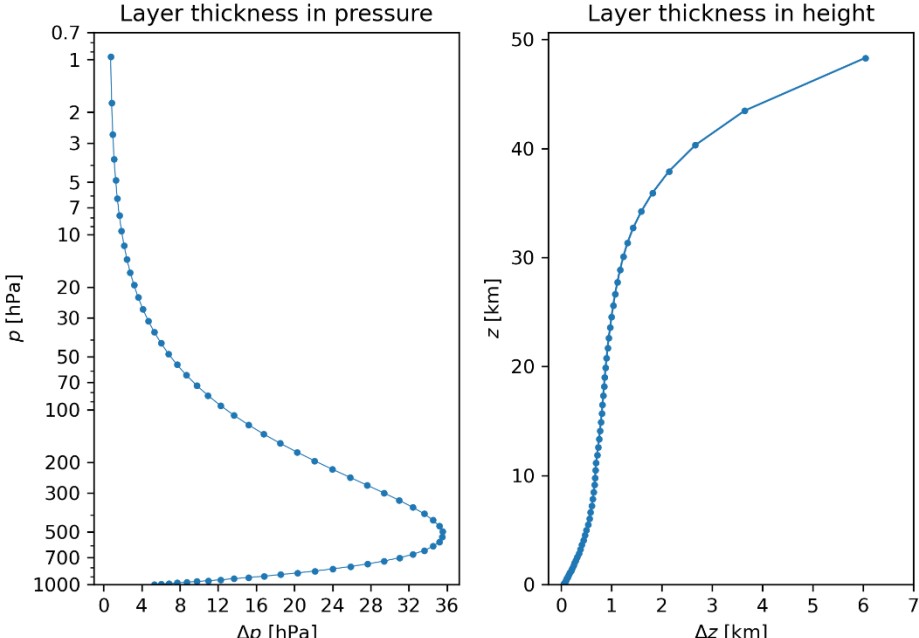

**Figure 2: Vertical levels used in simulations. Layer thickness in pressure $\Delta p$/height $\Delta z$ as a function of pressure $p$/height $z$ for a U.S. standard atmospheric with a surface pressure of 1000 hPa.**







| Configuration | Regional run | Global-nest run |
|---|---|---|
| Resolution | 37.5 km | 100 km/37.5 km |
| Acoustic timestep | 75 s | 150 s/75 s |
| Physics timestep | 450 s | |
| Microphysics | In-line GFDL microphysics (Harris et al., 2020) | |
| PBL scheme | Yonsei University scheme (Hong, 2010) | |
| Convection scheme | Scale-aware simplified Arakawa-Schubert (Han et al., 2017) | |
| Ocean surface | Mixed-layer ocean (Pollard et al., 1973) | |

Table 1: Model configurations of the regional and the global-nest run. For the global-nest run, resolution and acoustic timestep are
shown in a format of global/nest.

The containerized SHiELD has been tested on a variety of different x86 systems. Here we choose three machines – a laptop, a desktop, and a supercomputer – to demonstrate the portability, reproducibility, and performance of the containerized SHiELD. The three machines are chosen for their different OSs on different hardware, which are built for
different purposes, as listed in Table 2. They should be representative of computers used in most situations, including research, operation, and education.

| Machine | OS | CPU | Total cores |
|---|---|---|---|
| Laptop | Windows 10 | Intel Core i7-8550U | 4 |
| Desktop | Windows 10 | AMD Ryzen 9 3950X | 16 |
| Supercomputer | CentOS 7.6 | Intel Xeon Gold 6148 | 72000 |

Table 2: The operating system (OS), the central processing unit (CPU), and the total number of cores on the three representative machines.

All machines use the same Docker image to execute the containerized SHiELD for both regional and global-nest simulations. For the supercomputer, the Docker image is converted to a Singularity image for containerization, as described in Sec. 2.2. All machines use the same initial conditions and model configurations (and boundary conditions if the simulation is a regional run) to run a 24-hour simulation of Hurricane Laura. In either the regional or the global-nest case, all three machines finish the simulations successfully and give bitwise identical output even though they use different OS and
hardware. The containerized SHiELD has demonstrated its high portability and perfect reproducibility on any computing system that supports containerizing Docker images.



## 4 Result and Discussion

### 4.1 Hurricane Laura hindcast

Hurricane Laura was the first major hurricane (Category 4) that made landfall in the record-breaking 2020 Atlantic hurricane
season. Originating from a large tropical wave off the west African coast on August 16, Laura became a tropical depression
on August 20 and made landfall in the U.S. state of Louisiana early on August 27 after a period of rapid intensification.

Fig. 3 shows the simulated outgoing longwave radiation with both the regional and the global-nest configuration of
SHiELD, together with a satellite image of Laura for simple verification. At 12 hours into the integration, both simulations
capture the location and general structure of Laura well, even at a grid spacing of ~36 km. Laura is moving north toward
coastline. From the spiral rainbands, the size and the shape of the simulated Laura agree very well with the observation. Both
configurations give realistic simulations, demonstrating SHiELD as a truly unified modeling system. Although requiring
higher computational resources, one advantage of the global-nest run is that it allows two-way interaction between the nest
domain and the large-scale circulation in the global domain. Another advantage is that the global-nest run does not need a
time-series of boundary conditions, which could potentially introduce significant errors for a regional model, as discussed in
Warner et al. (1997). Comparison of the results suggests that this two-way interaction contributes to slightly faster
movement of the storm and brings it closer to the observation. Why Laura moves faster in the global-nest run is beyond the
scope of this work. At a relatively coarse resolution for hurricane simulation, this computationally inexpensive case study
serves to illustrate the capability of the containerized SHiELD.



Figure 3 image (a), (b), (c)

**Figure 3: Hurricane Laura simulations versus satellite imagery. Panel a and b show outgoing longwave radiation simulated by SHiELD as a region and a global-nest model, respectively. Both are a snapshot at 2020082700 UTC, 12 hours into the simulation. Dot box represents the boundaries of the nest domain. Panel c shows the Geostationary Operational Environmental Satellite-16 true color and night infrared imagery 10 minutes after 2020082700 UTC.**

## 4.2 Performance and scalability

Table 3 shows the wall-clock timings of the containerized SHiELD for the Hurricane Laura simulations on the three machines listed in Table 2 with the laptop as the baseline. In the regional case, each machine runs a 24-hour simulation using 2 Message Passing Interface (MPI) processes. The results reflect the per-core performance on each system with the desktop being the fastest, followed by the supercomputer and the laptop. In the global-nest case, each machine runs a 24-hour simulation using 8 MPI processes, 6 more than used in the regional case. The result shows the same ranking: 1) desktop, 2) supercomputer, and 3) laptop. However, this time the laptop runs over 100% more slowly than the other two systems. This





considerable slowdown is because the 8 MPI processes are oversubscribing only 4 cores (see Table 1). For the desktop and the supercomputer, both systems have more cores than required and therefore perform at comparable speeds, as it was with the regional case.


| Machine | Regional run | | Global-nest run | |
|---|---|---|---|---|
| | Wall time [s] | Cost relative to Laptop | Wall time [s] | Cost relative to Laptop |
| Laptop | 739.80 | 1.00 | 1889.66 | 1.00 |
| Desktop | 526.80 | 0.71 | 778.81 | 0.41 |
| Supercomputer | 598.98 | 0.81 | 808.05 | 0.43 |

**Table 3: Wall time of the regional and the global-nest run for a 24-hour simulation of Hurricane Laura on different machines. The regional run uses 2 MPI (Message Passing Interface) processes and the global-nest run uses 8 MPI processes.**

We have shown that the containerized SHiELD delivers reasonable performance that reflects the power of the hardware, regardless of the type of OS on the host machine. The next question to answer is how well the containerized SHiELD scales.

For scalability purposes, the 24-hour global-nest case discussed previously will be executed with differing numbers of cores on the supercomputer and compared against SHiELD running natively (i.e., without containerization). Fig. 4 shows the wall-clock timings for runs from 8 to 64 cores. Both the container and the native SHiELD give reasonable scalability. Interestingly, the containerized SHiELD runs about 10% faster than the native one at the same number of cores when using equal to or less than 32 cores. It is unclear why the containerized SHiELD outperforms the native one in some situations in

spite of the overhead introduced by the software container, albeit tiny. It could be due to the difference in the runtime environment between the Docker image and the supercomputer, which makes the executable in the Docker image more efficient. Regardless, this result shows that the use of a software container does not appreciably impact application performance, which is consistent with the findings of Felter et al. (2015) where the Docker container introduces "negligible overhead for CPU and memory performance".

When using 64 cores, the situation is the opposite: the native SHiELD outperforms the containerized version. Since each node on the supercomputer has 40 cores, this result could imply that the scalability of the container degrades across nodes. However, this is not the only explanation as the global-nest configurations can suffer from inherent load-imbalance. The global and nest domains run concurrently, rather than sequentially. The concurrent approach greatly improves computational efficiency, as discussed in Harris and Lin (2013), but it is not always possible to exactly synchronize the time per integration

step on the global and nest domains to avoid waits for one side or the other at the two-way update communication points. The current configuration uses a ratio of 3 to 1 to assign the number of MPI processes to the global and the nest domains. This configuration may not be optimal for the load balancing and therefore may not be suitable for scalability tests. Also, the problem size of the global-nest run may be too small to reveal the true scalability SHiELD can deliver.



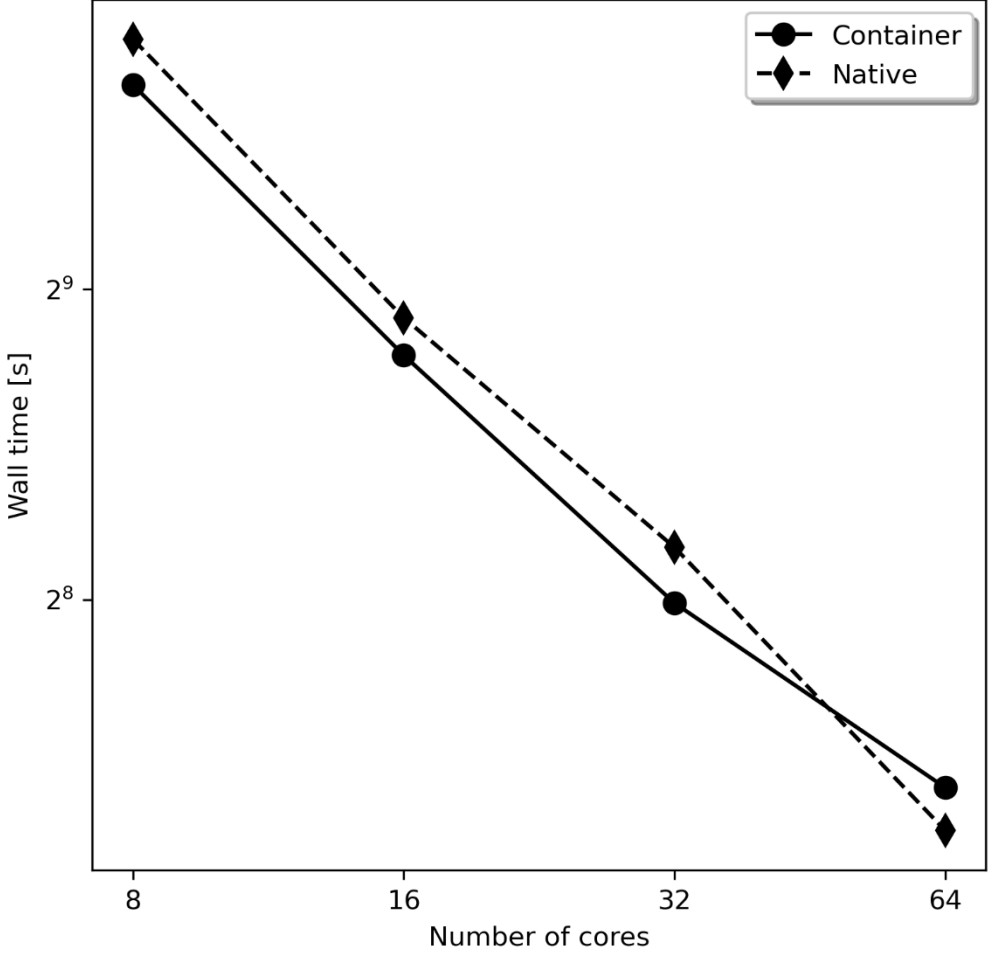

**Figure 4: Performance and scalability of the containerized SHiELD, compared to those of the native SHiELD. Results are from 24-hour global-nest simulations.**

### 4.3 Scalability issue

To better evaluate the scalability of containerized SHiELD, 12-hour global simulations at 13-km horizontal resolution were conducted. Unlike a global-nest run, the scalability test for a global run is straightforward in terms of MPI configurations. The global domain (cubed-sphere grid) is engineered to achieve excellent scalability, as demonstrated during the dynamical core evaluation of the Next-Generation Global Prediction System ( https://www.weather.gov/sti/stimodeling_nggps_implementation_atmdynamics, last access: 13 July 2021.). The global run
conducted here uses configurations similar to those used by the flagship 13-km SHiELD, the details of which can be found in Harris et al. (2020) or Zhou et al. (2019). Fig. 5a shows the data for runs from 96 to 3072 cores. The native SHiELD gives



good scalability, with speed roughly doubling as the number of cores used doubles. The containerized SHiELD, on the other hand, cannot scale beyond 768 cores, due to a large degradation in the MPI communication (Fig. 5b). When the number of cores used is equal to or less than 768 cores, the containerized SHiELD outperforms the native one, which is consistent with the result seen in the global-nest case (Fig. 4). However, the performance gap decreases with increasing number of cores. This is due to a disproportionate increase in the MPI communication burden. As shown in Fig. 5b, the MPI communication efficiency degrades with increasing numbers of cores for both containerized and native SHiELD. However, the degradation grows more quickly with the container. In the container case, the degradation becomes even worse when the simulation uses more than 768 cores. This result, combined with the results previously presented for the global-nest run, suggests the containerized SHiELD does scale reasonably well up to a modest number of cores, but the extreme scalability is significantly worse than that of the native SHiELD with an eventual plateau in performance. The modest core number seems proportional to the size of the problem, but an exact relationship is still unclear.

The MPI communication of the Singularity container remains challenging when an application uses a large number of cores. Note that, in addition to the "hybrid model" adopted in this study, the "bind model" provided by Singularity is likely to alleviate the scalability issue (https://sylabs.io/guides/3.7/user-guide/mpi.html, last access: 13 July 2021). The bind model uses the MPI implementation on the host to run a container, which should make the container fully compatible with the high-performance interconnects on the host. If the containerized SHiELD is built using the bind model, it may not show the scalability limit as seen in Fig. 4 and 5. However, the configuration of the bind model can be more complicated than the native SHiELD installation, as users are required to install all the libraries needed by SHiELD on the host and to bind them to the container explicitly. Also, reproducibility is not guaranteed with the bind model since the runtime environment of the bind model is dependent on the host and is not isolated as in the case of the hybrid model. For large-scale applications in which scalability is important, we recommend a native installation of SHiELD built from source.

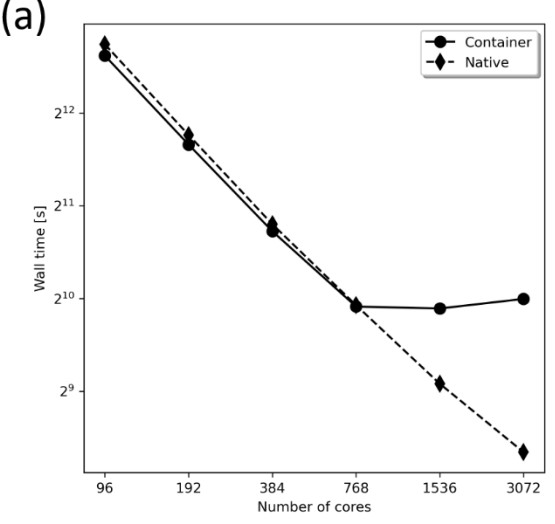
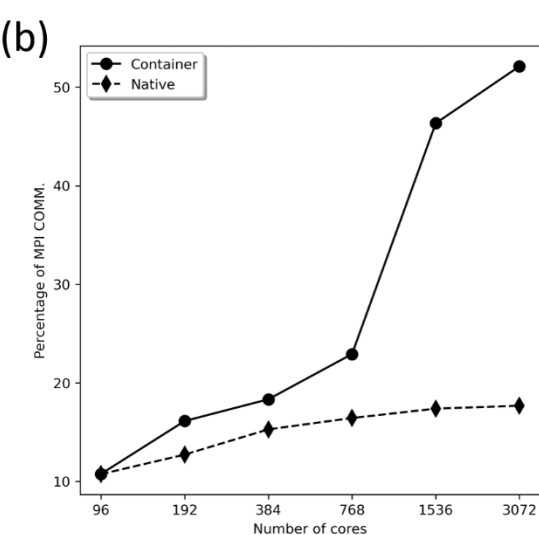



**Figure 5: (a) Performance and scalability of the containerized and the native SHiELD; (b) Ratio of the MPI communication to the**
**total computational cost in the case of the containerized and the native SHiELD. Results are from 12-hour global simulations at a**
**resolution of ~13 km.**

## 5 Conclusion and Future work

We have shown that software containers are a promising tool to enhance the accessibility of unified modeling systems,
which opens possibilities for research, education, and operation. We documented the implementation of software containers
for SHiELD, a compact unified modeling system developed at GFDL. To assure wide access and encourage collaboration,
the containerization of SHiELD was made without any proprietary software. Furthermore, the containerization described in
this study should be directly applicable to all unified modeling systems other than the one used here. We believe that making
unified modeling systems approachable will greatly accelerate model development.

The containerized SHiELD has demonstrated its high portability without compromise on functionality. It is fully
functional as the native one and can be easily deployed onto any computing system, as long as the system supports
containerization of Docker images. Simulations of Hurricane Laura with a regional and a global-nest configuration were
conducted on a variety of different x86 systems to show the flexibility of the containerized SHiELD. In either configuration,
bit-for-bit reproducibility is achieved regardless of differences in computing systems. The high portability and perfect
reproducibility brought by software containers enable reproducible research and analysis.

We demonstrated that the SHiELD container can be deployed on supercomputers across nodes using Singularity. The
container scales well up to a certain number of cores, depending on the size of the simulations. Beyond that certain number
of cores, the MPI communication burden grows quickly and degrades scalability drastically. The scalability issue of the
container could be solved by utilizing the bind model of Singularity. However, the bind model is nearly as difficult to
configure as the native SHiELD and does not guarantee reproducible results. A native installation of SHiELD is
recommended if scalability is critical and portability is less important.

The containerized SHiELD is designed to be community-oriented. We will continue to bring new features to the SHiELD
container using developments made within the experimental SHiELD. Efforts are being made to improve physics-dynamics
coupling by taking advantage of the conservation laws upon which the FV3 dynamical core is built. New capabilities (for
example, multiple and telescoping nest domains) are also being developed or planned. Meanwhile, we are working on
enabling idealized experiments, such as the Held-Suarez test (Held and Suarez, 1994) and supercell simulations. We believe
that these new capabilities will be useful for research or classroom learning.

**Code and data availability**

The containerized SHiELD developed in this study is available at https://zenodo.org/record/5090895 (Cheng et al., 2021). It is also available as a Docker image at https://hub.docker.com/r/gfdlfv3/shield tag gmd2021. The case configurations and associated initial/boundary conditions are available at https://zenodo.org/record/5090124 (Cheng, 2021b). The Laura simulations are available at https://zenodo.org/record/5090126 (Cheng, 2021a). The Geostationary Operational Environmental Satellite-16 data is available at the NOAA Comprehensive Large Array-data Stewardship System (CLASS; https://www.avl.class.noaa.gov/, last access: 13 July 2021).

**Author contribution**

KYC and LMH established the scientific scope of this study. KYC developed the containerization of SHiELD. KYC designed the experiments, conducted the simulations, and analyzed the model performance and scalability. YQS carried out the verification and analysis of Hurricane Laura simulations. KYC and YQS drafted the paper. All authors contributed to the writing of the finalized paper.

**Competing interests**

The authors declare that they have no conflict of interest.

**Acknowledgments**

We thank Linjiong Zhou for helping plot Fig. 2; Jeremy McGibbon and Spencer Clark for providing introduction of Docker and container technology; Rusty Benson and Thomas Robinson for providing reviews of this paper. This study is supported under awards NA18OAR4320123, NA19OAR0220145, and NA19OAR0220146 from the National Oceanic and Atmospheric Administration, U.S. Department of Commerce. The statements, findings, conclusions, and recommendations are those of the authors and do not necessarily reflect the views of the National Oceanic and Atmospheric Administration, or the U.S. Department of Commerce.

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
