# Peer review of "Enhancing the accessibility of unified modeling systems: GFDL SHiELD v2021b in a container"

_Geoscientific Model Development, 2021_

## Author Response (AR1)

We would like to thank both reviewers for their constructive comments of our paper. The following are specific responses to the suggestions given or questions raised by them. Original comments are in italic and responses are in regular font.

Reviewer #1:

1. *I would have like to see them go a bit further with the Singularity container, and in particular test the bind model.*

   We have added the results of the bind model and discussed its advantages and disadvantages in the revision. (Figure 5; L246-256; L273-275)

2. *The authors should carefully read the manuscript and augment every paragraph and figure caption to be explicit about whether it's about a Docker or Singularity container.*

   Done. (L184-185; L209; L235) Thank you.

Reviewer #2:

1. *P2, l39: "A software container, or simply container, is a stand-alone, executable software ..." -> This is not correct, the container is a software artifact that can be instantiated (in an image) and the runtime can "run them". This is better explained later on in the paper (e.g. P3, l38). Could you please rephrase this?*

   Done. (L39) Thank you.

2. *P3, l71: "As a result, a container is lightweight and fast." -> This is not necessarily true, containers can be big and problematic to build or run. In fact, your image is 576MB which is far from lightweight. I believe that if you want to mention one benefit of using containers (that is aligned with your work) it must be "portability". Could you please rephrase this?*

   What we wanted to say is that "a container is lightweight and fast, **compared to a virtual machine**". We have revised it to improve clarity (L70). Regarding the image size, it would be difficult to reduce the size substantially as the base image (centos 8) is far from slim and the GCC libraries are needed despite their size. We tried using slim base images (e.g., alpine) but the performance was disappointing (at least 10% slower). In the end, we decided to sacrifice image size for better performance.

3. *P3, l90: Recently, Docker has changed their licensing (https://www.docker.com/pricing/faq) and this could be problematic for reproducibility (which is one of the main topics of your work),*

*I suggest mentioning this in the paper and some open alternatives (such as Podman, https://podman.io/).*

Done (L90-93). Thank you.

4. *P10: I think you can find interesting (and very related to this section) the paper: "Montes D, Añel JA, Wallom DCH, Uhe P, Caderno PV, Pena TF. Cloud Computing for Climate Modelling: Evaluation, Challenges, and Benefits. Computers. 2020; 9(2):52. https://doi.org/10.3390/computers9020052 "*

Thank you for providing this interesting paper. We have addressed the scalability issues related to external hardware/software and cited related papers (L246-256).

5. *P12: Open question (that I believe can add more value to the conclusions): Can the model run in more than one container? If so, what would be the potential benefits of running it on an orchestration system such as Kubernetes or AWS ECS?*

Thank you for this thought-provoking question. We have no problem running multiple SHiELD containers at the same time. We think that an orchestration system will be useful for the SHiELD container in terms of resource management. An orchestration system can automate the workflow of conducting a simulation across different computing infrastructures, which will be useful for ensemble forecast/hindcast. The idea is attractive especially because an orchestration system is able to balance workload and distribute multiple runs so that the whole ensemble forecast/hindcast becomes stable and efficient. However, while the application of an orchestration system on the SHiELD container is interesting, for this study, we want to keep the focus on the SHiELD container itself. Using an orchestration system to develop a SHiELD ensemble system is something we may work on in the future.

6. *Is there any public registry with an available SHiELD image that can be consumed? Building the image every time is a time-consuming task (>40 minutes for your case, on a high-end machine) and could also lead to a potential issue with reproducibility (e.g. broken dependency). If so, could you please link it to the paper?*

Yes, the SHiELD image is publicly available on Docker Hub at https://hub.docker.com/r/gfdlfv3/shield tag gmd2021. This link can be found in the section "Code and data availability", as well as the asset "Executable research compendia (ERC)".

**List of changes:**

L32: updated date

L39: revised in response to Comment 1 by Reviewer #2

L64: added "this"

L71: revised in response to Comment 2 by Reviewer #2

L84-85: updated date

L90-93: revised in response to Comment 3 by Reviewer #2

L98: updated month

L105: corrected date

L184-185: revised in response to Comment 2 by Reviewer #1

Figure 4: replotted with color

L209: revised in response to Comment 2 by Reviewer #1

L217: updated date

Figure 5: added results of the bind model

L235: revised in response to Comment 2 by Reviewer #1

L236-245: added a paragraph in response to Comment 4 by Reviewer #2

L246-256: revised in response to Comment 1 by Reviewer #1

L273-275: revised in response to Comment 1 by Reviewer #1

L288: updated date

L298: added two anonymous reviewers

L344: added citation

L361: added citation